

# Optimizing carbon nanoparticle staining for sentinel lymph nodes in rabbit lower limb models: concentration and time dependence

Narbol Kylyshbek[1,2,*], Heng Wang[1,2,*], Shuning Feng[1,2], Yuxin Zhang[1,2], Dong Dong[1,2], Wangjaleoseop Han[1,2], Ayidana Ayoujiang[1,2], Liang Chen[1,2] and Tianyi Liu[1,2]

[1] Shanghai Key Laboratory of Clinical Geriatric Medicine, Huadong Hospital, Fudan University, Shanghai, China

[2] Department of Plastic and Aesthetic Surgery, Huadong Hospital, Shanghai Medical College, Fudan University, Shanghai, China

[*] These authors contributed equally to this work.

## ABSTRACT

**Background and Objectives**. Carbon nanoparticles (CNPs) have been used as a lymphatic imaging agent that can selectively enter lymphatic vessels. This experiment used CNPs for lower limb lymph node staining in rabbits, aiming to explore the suitable concentration and time of CNPs for sentinel lymph node (SLN) sufficient staining, and the biosafety of CNPs retention was preliminarily explored.

**Methods**. Staining effect of SLN using different concentrations of CNP injection on the rabbit lower limb lymphatic model. To explore the temporal correlation of CNPs staining. We investigated the effects of long-term retention of CNPs in lymph nodes on lymph node tissue structure and mRNA expression of inflammatory factors (IL-6/TNF-$\alpha$), apoptosis factors (Bcl-2/Bax), and fibrosis factors (Collagen-I/$\alpha$-SMA).

**Results**. The lowest concentration to achieve adequate lymph node staining is 0.0781 mg/0.8 mL, while the best staining results were observed at least 12 hours post-injection. Long-term CNPs retention has no significant negative impact on the structure and function of lymph nodes.

**Conclusions**. CNPs are an effective and safe lymphatic tracer. It is a concentration-dependent and time-dependent lymphatic staining agent. 0.0781 mg/0.8 mL and 12 hours are suitable for the rabbit lower limb lymphatic staining model, which provides valuable data for further research on the application of CNPs in rabbit animal models. In addition, the results provide evidence for future clinical development of concentration and time standards for CNPs' use.

# INTRODUCTION

Sentinel lymph node biopsy (SLNB) now is a common technique used in the treatment of cancer, helping to advance the creation and application of less invasive surgical techniques

Corresponding author
Tianyi Liu, tianyiliuhd@163.com

(*Giammarile et al., 2022*). Furthermore, SLNB is essential to identifying latent micro-metastases in lymph nodes that drain a particular region. Disease staging, disease-free survival, and overall survival are all significantly predicted by the presence of malignant tumor cells in the sentinel lymph node (SLN) (*Arnold et al., 2022*; *MacKie, Hauschild & Eggermont, 2009*). SLNB has no detrimental effects on disease-free survival or overall survival rates and is a less intrusive procedure. Additionally, compared to lymph node dissection, SLNB had decreased incidence of lymphedema and other postoperative morbidities (*Zhai et al., 2021*; *Abu-Rustum, 2014*).

Tracing the lymph nodes is crucial for precise SLNB. At present, common tracers for SLN include radioactive substances, blue dyes, fluorescent dyes, and carbon nanoparticles (CNPs). Among them, blue dye has low sensitivity, poor visibility, and can cause 0.1% to 2.7% of adverse reactions, such as skin staining and allergic reactions. Fluorescent dyes, such as indocyanine green (ICG), provide better tissue penetration and visualization compared to blue dyes, but require additional specialized equipment assistance (*Van den Berg et al., 2014*; *KleinJan et al., 2016*; *Xie et al., 2017a*). CNPs are the only new carbon nanomaterials that have been marketed and used in therapeutic settings (*Frontado et al., 2013*). CNPs are composed of nanoscale carbon particles with an average particle size of 150 nm. It is a novel nanomaterial lymphatic tracer that can selectively enter lymphatic vessels without entering blood vessels, achieving targeted staining of the lymphatic system. Unlike radioactive tracers and fluorescent dyes, CNPs does not require specialized equipment, making it suitable for hospitals with limited medical conditions. Compared with other dye lymphatic tracers, CNPs seems to have more advantages in SLN identification and lymphatic vessel visualization. By accurately identifying lymph nodes, they can also shorten surgical times and lower the risk of problems following surgery (*Sun, Bai & Wang, 2023*). Currently, CNPs is widely used in SLN of thyroid cancer, gastric cancer, colorectal cancer and breast cancer, showing good lymph node recognition rate and biosafety (*Zhao et al., 2017*; *Gershenwald, 1999*; *Li et al., 2016*; *Wang et al., 2017*; *Zhang et al., 2018*).

Based on a comprehensive review of previous literature, this is the first study to systematically quantify both the optimal concentration and time dependence of carbon nanoparticle staining in a rabbit lower limb lymphatic model. Moreover, our work extends beyond imaging optimization by evaluating the long-term biosafety of CNPs through both histological and molecular (RT-qPCR) analyses. These dual-level safety assessments offer novel insights into the compatibility of CNPs with lymphatic tissue, providing a more comprehensive reference for their potential clinical use.

Although CNPs has been widely used in clinical practice, there is no unified standard for the injection dose of CNPs and the exploration time of SLN after injection, which prompted us to conduct this study. Despite the widespread use of CNPs in clinical practice, the market price of CNPs is relatively expensive, and the cost of using it for patients is high. Under the premise of fully staining lymph nodes, injecting CNPs at an appropriate concentration will reduce the dosage of CNPs used, which will help lower the treatment cost for patients and reduce the likelihood of adverse reactions caused by excessive CNPs injection. In this study, we used New Zealand rabbits lower limb lymph staining model to preliminarily investigated the concentration of nanocarbon used when lymph nodes were

fully stained. Explored the temporal correlation of nanocarbon staining. Studied the effects of prolonged retention of nanocarbon in lymph nodes on lymph node tissue structure and mRNA expression of inflammatory, necrotic, and fibrotic factors. These findings provide reference for the application of CNPs in animal models and lay a certain foundation for further standardization of clinical CNPs application.

## MATERIALS & METHODS

### Carbon Nanoparticle Suspensions (CNPs) preparation

A one mL syringe was used to withdraw 0.1 mL of a five mg/mL CNPs. Following this, 0.7 mL of physiological saline was added to the syringe, and 0.4 mL of the resulting mixture was retained for storage. Subsequently, an additional 0.4 mL of physiological saline was drawn into the syringe. At this stage, the concentration of the suspension was 2.5 mg/mL. The above steps should be repeated to achieve the desired concentrations for different experimental groups (0.3125 mg/0.8 mL, 0.1562 mg/0.8 mL, 0.0781 mg/0.8 mL, 0.0390 mg/0.8 mL, 0.0195 mg/0.8 mL, 0.0097 mg/0.8 mL.)

### Animals and injection procedures

The New Zealand White rabbits (2.5–3 kg) were housed in standard laboratory conditions with ad libitum access to food and water. Enrichment materials, such as chew toys and bedding, were provided to promote natural behaviors. Twenty-four rabbits were randomly assigned to six groups for dose exploration, receiving varying doses of CNPs (0.3125 mg/0.8 mL, 0.15625 mg/0.8 mL, 0.078125 mg/0.8 mL, 0.0390 mg/0.8 mL, 0.0195 mg/0.8 mL, and 0.0097 mg/0.8 mL) *via* a four-point footpad injection. Another group of eight rabbits received a standard dose (0.078125 mg/0.8 mL) to assess the temporal effects at 4, 8, 12, and 16 h post-injection. All animals received meloxicam (1 mg/kg, subcutaneously) 30 min before the procedure and buprenorphine (0.05 mg/kg, subcutaneously) after injection, repeated as necessary to minimize pain and distress. Euthanasia was performed at the end of the experiment *via* intravenous sodium pentobarbital (100 mg/kg). Criteria for premature euthanasia included signs of severe distress or irreversible harm, but none of the animals required early euthanasia. Surviving animals were euthanized, and tissues were collected for histological analysis, with carcasses disposed of according to institutional protocols. Ethical approval has been obtained from the Institutional Animal Care and Use Committee (IACUC) under the approval number IACUc-2023-Ra-009. Shanghai Yishang Biotechnology Co., Ltd. All experimental procedures adhere to the guidelines for animal welfare and ethical standards in scientific research.

### CNPs Injection and SLNB acquisition

Sodium pentobarbital is used for anesthetizing the rabbits and fix their lower limbs, after depilating the hind footpads of rabbits, CNPs were injected subcutaneously using a four-point method (3, 6, 9, and 12 o'clock). After waiting for the target time, The rabbit was euthanized by intravenous air injection *via* the ear vein following deep general anesthesia, during which the animal showed no signs of pain or struggle. The popliteal area was

depilated, locally disinfected, and a longitudinal skin incision of about one cm was made on the popliteal skin. The subcutaneous soft tissue was carefully dissected and separated, and the stained lymph nodes were found. The blood vessels around the lymph nodes and the inflow and outflow lymph vessels were cut off, and the lymph nodes were removed for further testing.

## The effects of prolonged retention of CNPs in SLNB

To investigate the effects of prolonged retention (2 weeks after injection) of CNPs in lymph nodes, the histology, immunofluorescence staining, and RT-qPCR were performed. For H&E staining, sections were deparaffinized, hydrated, stained with hematoxylin and eosin, dehydrated, cleared, mounted, and finally examined under a microscope for image analysis. For immunofluorescence staining, tissue sections were first deparaffinized by incubation in xylene for 15 min, repeated three times, followed by rehydration through sequential immersion in 100% ethanol, 95% ethanol, and 70% ethanol for 5 min each. The sections were then incubated with primary antibodies against LYVE-1 (HZ-20120R; Shanghai Huzhen Industrial Co., Ltd, Shanghai, China) and CD31 (HZ-0195R; Shanghai Huzhen Industrial Co., Ltd, Shanghai, China). This was followed by incubation with HRP-conjugated secondary antibodies (rabbit anti-goat HRP, bs-0296R; Bioss) and FITC-conjugated secondary antibodies (goat anti-rabbit FITC, bs-0295G; Bioss). Lymphatic vessel density was estimated based on five hotspots at high magnification, and CD31-marked vessel density was counted. Total RNA was extracted from lymph node tissues using Trizol (#R0016; Beyotime, Shanghai, China) according to the manufacturer's instructions. RNA concentration and purity were assessed using a NanoDrop spectrophotometer (Thermo Fisher Scientific). cDNA was synthesized from one μg of total RNA using the BeyoRT™ III First Strand cDNA Synthesis Kit (#D7178M; Beyotime) in a 20 μL reaction volume, following the manufacturer's protocol. Quantitative PCR (qPCR) was performed by the LightCycler® 480 Instrument II. Gene expression of inflammatory factors (IL-6/TNF-α), apoptosis markers (Bcl-2/Bax), and fibrosis factors (Collagen-I/α-SMA) were analyzed. The mRNA levels were normalized to b-actin and analyzed using the $2^{-\Delta\Delta T}$ method (*Pfaffl, 2001*). GAPDH served as the internal reference gene for calibrating gene expression levels. The primers used are listed in Table 1.

## Image and statistics

Image-Pro Plus 6.0 Software was used to analyze the images. Statistical analyses were performed using GraphPad Prism v8.0.2.263. Statistical significance and $p$ values are analyzed by one-way ANOVA followed by Tukey's multiple comparisons test. Cell counts of LYVE-1/CD31 and gene expression levels were analyzed using $t$-tests. Results are presented as mean values ± SD. And where $^\star p < 0.05$ indicates statistical significance.

# RESULTS

## Effective staining in rabbit lower limb lymphatic staining model

The effect of concentration of carbon nanoparticles on staining effectiveness was examined. Rabbits were given subcutaneous injections of increasing doses of CNPs to examine the

**Table 1  GAPDH served as the internal reference gene for calibrating gene expression levels.** The primers used are listed in this table.

| Gene name | Upstream sequence | Downstream sequence |
|---|---|---|
| IL-6 | ATGGATGCTACCAAACTGGAT | TGGTACTCCAGAAGACCAGAG |
| TNF-a | ATGGCCTCCCTCTCATCAGTTC | AGATAGCAAATCGGCTGACGG |
| Bcl-2 | ATGGGAGAACAGGGTGGACA | CTACCGTGGGCTCCATAGAC |
| BAX | ATGGCGTCCACCAAGAAGCTG | GTCAGCTGCCGGTTTCAAGA |
| a-SMA | ATGGTGATGGACTCCGGAGA | TTGGTGATGATGCCGTGTTG |
| Collagen-I | ATGGCTTCGGCTGTGGTCTG | GGGAGCCGGGTCCTCGTGGA |
| β-actin | CGGTTTGCCGAGAGGTT | CCTTCTGCATCCTGTCAGCA |

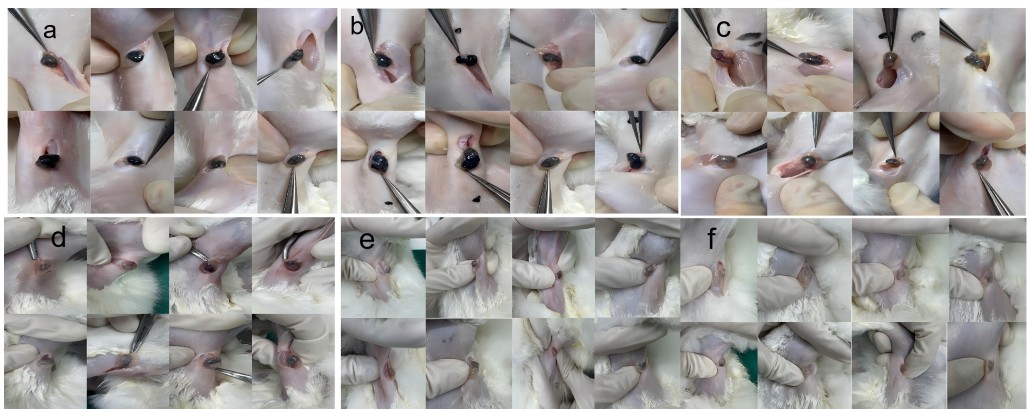

**Figure 1  Observation of popliteal lymph nodes in rabbits 12 h after injection of nano-carbon into the plantar region of the foot.** Staining intensity at different doses: (A) 0.3125 mg, (B) 0.15625 mg, (C) 0.078125 mg, (D) 0.039 mg, (E) 0.019 mg, (F) 0.0097 mg. The staining intensity of popliteal lymph nodes was assessed following injection of different doses of nano-carbon into the plantar region of the foot in rabbits after a 12-hour period.

impact of various CNP concentrations on lymph node staining. Both *in vivo* and *ex vivo* views of lymph node staining were detected (Figs. 1 and 2). The results of Image-Pro Plus 6.0 analysis lymph node staining grayscale values show that the capacity of the lymph nodes to be stained enhanced in proportion to the rise in CNP dosage. CNPs outperformed lesser doses in staining efficacy ($p < 0.01$) at a concentration of 0.078125 mg (Fig. 2G). Nevertheless, additional concentration increases did not result in a statistically significant improvement in staining ($p > 0.05$), suggesting that 0.078125 mg is adequate for rabbit lymph node labeling.

## Staining efficiency improved with longer injection times

After confirming that 0.078125 mg/0.8 mL is the appropriate concentration for rabbit lymph node imaging, CNPs at this dose were used to explore the impact of different staining times (4, 8, 12, and 16 h) on lymph node staining effectiveness. Both *in vivo* and *ex vivo* observations confirmed that staining efficiency improved with longer injection times ($p < 0.05$) (Fig. 3). According to the grayscale analysis, 12 h and 16 h were statistically

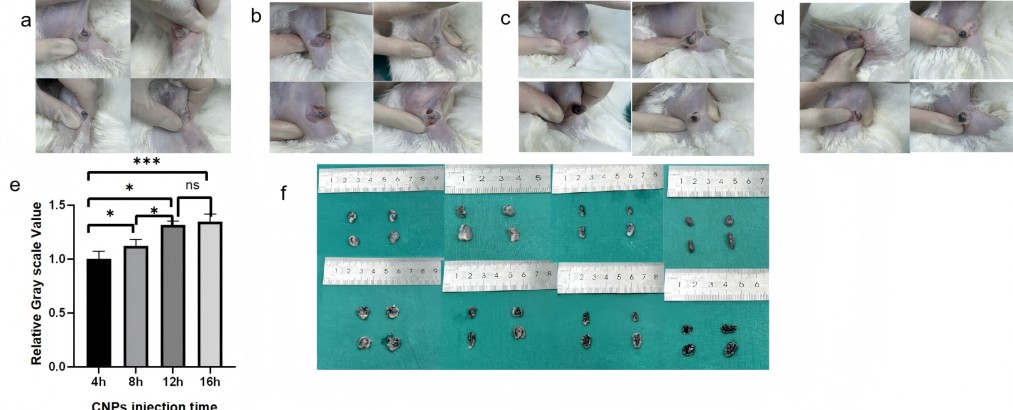

**Figure 2 Longitudinal cross-sectional images of popliteal lymph nodes extracted from rabbits at different doses.** (A) 0.3125 mg, (B) 0.15625 mg, (C) 0.078125 mg, (D) 0.039 mg, (E) 0.019 mg, (F) 0.0097 mg. Panel (G) depicts tissue grayscale analysis, showing a cliff-like decrease below 0.78125 mg. The longitudinal cross-sectional images of popliteal lymph nodes extracted from rabbits at varying doses are presented in panels (A) to (F), with grayscale analysis shown in panel (G) indicating a sharp decline below 0.078125 mg, as observed in a cliff-like fashion.

**Figure 3 Observation of staining in popliteal lymph nodes at a uniform concentration (0.078125 mg) following injection at different time points.** ((A) 4 h, (B) 8 h, (C) 12 h, (D) 16h). Subsequent longitudinal sectioning (F) and grayscale analysis (E) revealed an increasing trend in staining effectiveness with longer observation times. There was no statistical significance between the 12 h and 16 h groups ($p > 0.05$). The staining patterns in popliteal lymph nodes were monitored at a consistent concentration of 0.078125mg after injection at various time intervals (4 h, 8 h, 12 h, 16h). Following longitudinal sectioning (F) and grayscale analysis (E), it was observed that the staining effectiveness exhibited an upward trend with increasing time. Notably, there was no statistically significant difference between the 12 h and 16 h groups ($p > 0.05$), as indicated by the analysis.

insignificant ($p > 0.05$), proving that 12 h after injection of CNPs is the starting time for sufficient staining of sentinel lymph nodes.

## CNPs are safe and effective lymph node imaging agent

To investigate whether CNPs can cause adverse reactions in tissues, a series of tests were conducted for verification. After nanocarbon staining, HE staining of the lymph nodes

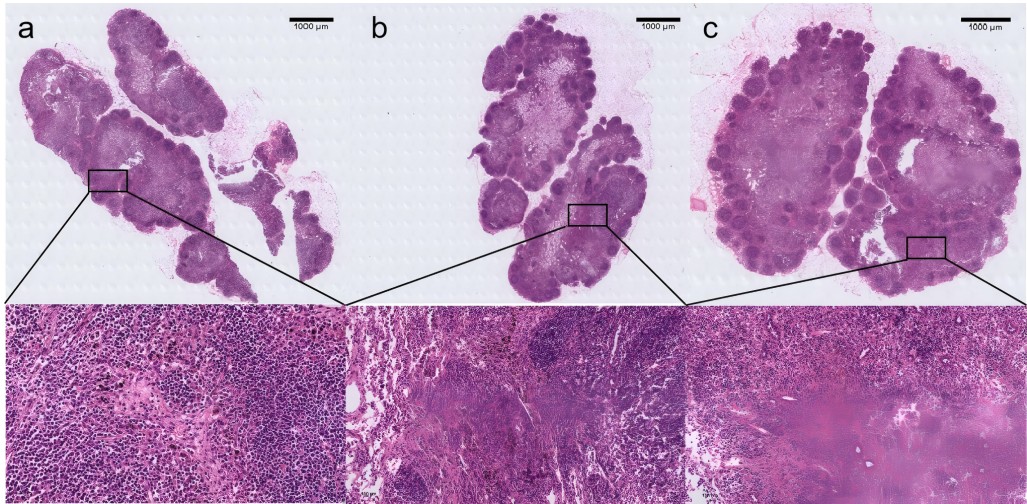

**Figure 4** **H&E-stained histological sections of rabbit lymph nodes showing typical lymph node architecture.** With distinct regions including the cortex, paracortex, and medulla. Histological analysis confirms the structural integrity of the lymph nodes, with clearly demarcated areas. In all experimental groups, tissues stained with CNPs were observed. The scale bar for the magnified image is 50 μm. Panels represent different experimental conditions: (A) dosage group, (B) time group, (C) 2-week carbon nanoparticle retention.

reveals that, in comparison to normal lymph nodes, the lymph nodes' shape is regular, their contour is complete, and the quantity and distribution of cells inside the lymph nodes are normal (Fig. 4). Use immunofluorescence to evaluate the pan vascular marker CD31 and lymph specific hyaluronic acid receptor LYVE-1 to assess the effect of CNPs on lymph node vasculature and lymphatic vessels. The results indicate that the prolonged retention of CNPs did not affect the number of blood vessels and lymphatic vessels within the lymph nodes (Fig. 5).

In addition, RT qPCR was used to evaluate the mRNA expression levels of inflammatory, necrotic, and fibrotic factors in lymph nodes. The results showed that long-term retention of CNPs did not increase the expression of cytokines related to lymph nodes ($p > 0.05$) (Fig. 5G), further confirming the biocompatibility of CNPs and its effectiveness as a safe lymph node imaging agent.

In addition, there is no staining agent leakage around the lymph nodes stained with carbon nanoparticles, as demonstrated in Fig. 2, which also shows how the black lymph nodes stained with nanocarbon have a dramatic color contrast with the surrounding tissues and can even be seen through the skin. Furthermore, after injecting CNPs, the rabbits' feeding, drinking, and resting patterns remained unchanged, and no inflammatory symptoms, such as swelling or rupture, were noticed at the injection site until the completion of the trial.

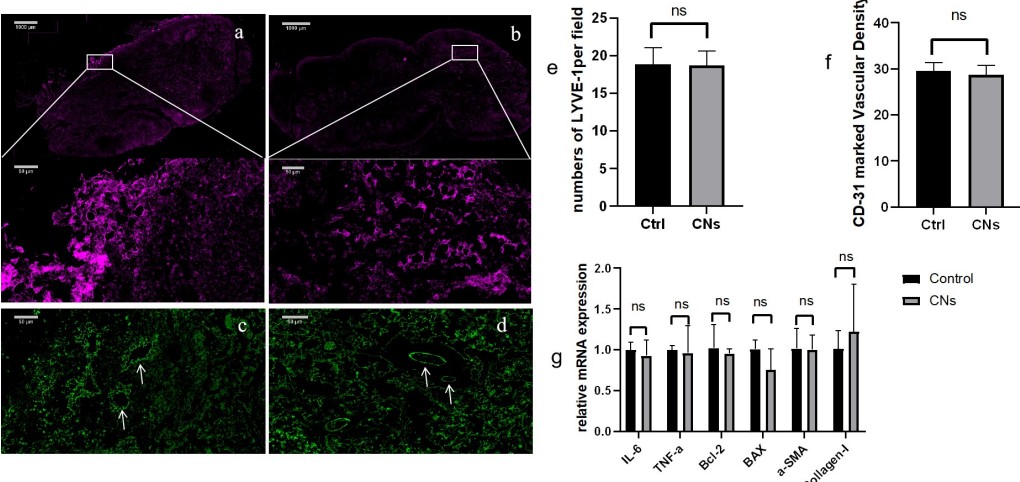

**Figure 5  The immunofluorescence results of the CNPs injection group (A, C) and the control group (B, D).** Low magnification images show bright staining for LYVE-1, high magnification images depict LYVE-1-labeled lymphatic vessels (A, B), and the distribution of CD-31 is observed at high magnification (C, D). The average LYVE-1 expression levels are shown in panel E, with no significant difference observed ($P > 0.05$). Panel H presents the average number of CD-31-labeled vessels, also showing no significant difference ($P > 0.05$). (F) Relative values were utilized for analysis in the q-PCR analysis against the control group, revealing no statistically significant differences across all groups ($p > 0.05$) (G).

# DISCUSSION

By performing preoperative and intraoperative lymphatic mapping and then selectively excising the first lymph node discovered in the lymphatic drainage channel leading from the main tumor to the regional nodal basin, a procedure known as SLNB is carried out. By identifying the lymph node most likely to have any cellular metastases from the original tumor, this procedure diagnoses clinically occult illness (*Morton et al., 1992*; *Morton et al., 1999*). CNPs have emerged as a promising lymph node tracer due to their unique properties, such as not relying on specialized equipment, ease of operation, and stable staining effects (*Khandker, Shakil & Hossen, 2020*). Following an intratumoral injection, CNPs can dye the tumor drainage lymph node black. Advanced stomach cancer, breast cancer, and papillary thyroid carcinoma have all seen successful demonstrations. Furthermore, CNP injections can significantly lower the risk of incorrect resection since they do not stain the parathyroid gland during thyroid cancer surgery (*Zhu et al., 2016*). CNPs demonstrated efficient lymph node staining and minimal side effects. This is mainly because CNPs can enter the lymphatic system efficiently and not enter the bloodstream, which lowers systemic toxicity (*Gu et al., 2015*; *Li et al., 2022*; *Chan et al., 2021*). Because CNPs don't require the use of radioactive isotopes like radioactive tracers do, there are fewer worries regarding radiation exposure. Research has indicated that while radioactive tracers such as Tc-99m are extremely sensitive in SLN biopsy, their radioactivity may put patients and medical personnel at risk (*Moncrieff & Thompson, 2022*). On the other hand, CNPs are just as effective at localizing lymph nodes but don't come with the same radiation safety concerns

(*Ottolino-Perry et al., 2021*). ICG provides high contrast imaging in near-infrared light and good tissue penetration when used as a fluorescent tracer (*Grosek & Tomažič, 2020*). But in environments with limited resources, ICG may not be possible due to the need for costly imaging equipment (*Rogach & Ogris, 2010*). In contrast, CNPs is a good substitute for ICG in settings with limited equipment because they don't need specific equipment and provide distinct staining effects (*Tian et al., 2022*). In this investigation, 0.078125 mg/0.8 mL was the suitable minimum concentration of CNPs in the rabbit lower limb lymph staining model, and at least 12 h after injection, a significant staining impact of the lymph nodes could be seen. Similar to other lymphatic dye tracers, the staining effect of nanocarbon exhibits concentration dependence and time dependence within a certain range. As shown in Fig. 2, the black lymph nodes stained with nanocarbon have a strong color contrast with the surrounding tissues, and even the black stained popliteal lymph nodes can be detected through the skin, making it easier to quickly locate the target lymph nodes and reducing surgical time; Moreover, there is no leakage of staining agents around the lymph nodes stained with CNPs, and the tissue can be clearly distinguished during dissection, which facilitates our precise surgical operation and further reduces the trauma caused by surgery and the incidence of related complications. In our experiment, no inflammatory reactions such as swelling or rupture were observed at the injection site of the rabbits until the end of the experiment, and after injecting CNPs, there was no change in daily habits.

In this study, we conducted tissue immunofluorescence analysis and RT-qPCR analysis on rabbit popliteal lymph nodes regarding the biological safety of long-term retention of CNPs. The experimental results showed that long-term retention of CNPs did not significantly affect the tissue structure of lymph nodes, and there was no differential increase in the mRNA expression of inflammatory factors(IL-6/TNF-α),apoptosis-related markers (Bcl-2/Bax), and fibrosis factors (Collagen-I/α-SMA). This is consistent with previous studies on the biocompatibility of nanomaterials, further supporting the potential of CNPs as a safe and effective lymph node tracer (*Fadeel & Garcia-Bennett, 2010*; *Xie et al., 2017b*). In addition, according to other animal experiments and clinical observations, CNPs has very low toxicity after exposure (*Debnath et al., 2025*). The endpoint time of our experiment is 2 weeks, and further exploration is needed to investigate the lymphatic system retention of nanocarbon for a longer period of time and whether it has adverse effects on various organs throughout the body.

In terms of timing for SLN staining, different tracers have shown varying optimal imaging times. For instance, blue lymphatic staining agents, generally, only need to be injected during surgery because of their small molecular weight and fast diffusion rate within the lymphatic system. 99mTc-labeled sulfur colloid typically achieves optimal imaging within 2 to 3 h post-injection (*Patel et al., 2021*). Our findings that CNPs achieve optimal staining at 12 h post-injection are consistent with current clinical practices, supporting the necessity of extending the staining period before surgery (*Jeremiasse et al., 2020*). The prolonged retention of CNPs in lymph nodes provides us with a wider surgical window for finding lymph nodes, but further research is needed to determine whether prolonged intervals after injection lead to more staining of non sentinel lymph nodes, especially in areas with complex lymph structures such as the axilla and groin. Nonetheless,

further research is needed to explore the long-term staining effects and the application of CNPs in other cancer types. Although our research indicates that a concentration of 0.078125 mg/0.8 mL of CNPs is effective for staining rabbit models, our experimental results provide some reference value in the relevant experiments of rabbit models.

Although our study determined the optimal minimal effective concentration and time for staining, future investigations should evaluate whether higher doses of CNPs could achieve sufficient lymph node staining in a shorter time frame. This approach could potentially benefit intraoperative SLN detection where time is critical. Further experimental groups exploring this relationship are warranted to expand the clinical applicability of CNPs. Our study was designed under experimental conditions aiming to optimize the staining parameters of CNPs in a preclinical rabbit model. This differs from intraoperative SLN detection in clinical settings, which emphasizes rapid visualization and real-time decision-making. The goal of this study is to determine the minimum effective concentration and appropriate staining time that ensure sufficient staining of sentinel lymph nodes in experimental conditions. These findings serve as a foundational reference for further translational studies and preoperative planning in clinical scenarios.

However, further research is needed for the application of nanocarbon in other animal models, as well as in clinical applications for different patients and disease types (*Gao et al., 2005*). Especially in cases involving more complex lymphatic systems or lymph node metastasis, higher or lower doses may be required to achieve optimal results (*Soares et al., 2018*). Recent studies have found that CNPs can also be used as drug carriers to load drugs for targeted therapy of tumor metastasis in the lymphatic system. They can reduce the concentration of drugs used and minimize side effects (*Huang et al., 2018*).

## CONCLUSIONS

In summary, our study found that CNPs is a concentration dependent and time-dependent lymphatic staining agent, and increasing the concentration and time of CNPs within a certain range can improve the staining effect of lymphatic formation. A total of 0.78125 mg/0.8 mL is suitable for the rabbit lower limb lymphatic staining model, which provides valuable data for further research on the application of CNPs in rabbit animal models. In addition, the safety of long-term retention of CNPs has also been confirmed in this experiment, and our research results provide evidence for future clinical development of concentration and time standards for CNPs use.

### Funding

This study was funded by the Medical Guidance Project of the Science and Technology Commission of Shanghai Municipality (22Y11905800) and Center for Scientific Research and Development, Ministry of Education (2021JH014). The funders had no role in study design, data collection and analysis, decision to publish, or preparation of the manuscript.

## Grant Disclosures

The following grant information was disclosed by the authors:

The Medical Guidance Project of the Science and Technology Commission of Shanghai Municipality: 22Y11905800.

Center for Scientific Research and Development, Ministry of Education: 2021JH014.

## Competing Interests

The authors declare there are no competing interests.

## Author Contributions

- Narbol Kylyshbek conceived and designed the experiments, performed the experiments, prepared figures and/or tables, authored or reviewed drafts of the article, and approved the final draft.
- Heng Wang conceived and designed the experiments, performed the experiments, prepared figures and/or tables, authored or reviewed drafts of the article, and approved the final draft.
- Shuning Feng conceived and designed the experiments, analyzed the data, prepared figures and/or tables, and approved the final draft.
- Yuxin Zhang analyzed the data, authored or reviewed drafts of the article, and approved the final draft.
- Dong Dong analyzed the data, prepared figures and/or tables, and approved the final draft.
- Wangjaleoseop Han analyzed the data, prepared figures and/or tables, authored or reviewed drafts of the article, and approved the final draft.
- Ayidana Ayoujiang analyzed the data, prepared figures and/or tables, authored or reviewed drafts of the article, and approved the final draft.
- Liang Chen performed the experiments, authored or reviewed drafts of the article, and approved the final draft.
- Tianyi Liu conceived and designed the experiments, prepared figures and/or tables, authored or reviewed drafts of the article, and approved the final draft.

## Animal Ethics

The following information was supplied relating to ethical approvals (*i.e.*, approving body and any reference numbers):

All procedures performed were evaluated and approved by the Shanghai Yishang Biotechnology Co., Ltd. (IACUC-2023-Ra-009).

## Data Availability

The raw measurements are available in the Supplementary Files.

## Supplemental Information

Supplemental information for this article can be found online at http://dx.doi.org/10.7717/peerj.19799#supplemental-information.

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
