# Peer review of "Optimizing carbon nanoparticle staining for sentinel lymph nodes in rabbit lower limb models: concentration and time dependence"

_PeerJ, doi:10.7717/peerj.19799_

## Round 0.1 · original submission · Major Revisions

**Language Note:** The review process has identified that the English language must be improved. PeerJ can provide language editing services - please contact us at [email protected] for pricing (be sure to provide your manuscript number and title). Alternatively, you should make your own arrangements to improve the language quality and provide details in your response letter. – PeerJ Staff

Reviewer 1 ·

Basic reporting

The authors identified the optimal amount of dye and staining time to identify lymph nodes in rabbit hind limbs. Further biocompatibility of the carbon nanoparticles after 14d was assessed by histology and RT-PCR detection of inflammatory cytokines and markers of apoptosis and fibrosis. The work does not mention that the requirements for the experimental detection of lymph nodes in this study and the selective intraoperative detection of lymph nodes are different. It is therefore necessary to specify the research question that the authors intend to address.

Experimental design

The experimental design overall is good. Bcl-2/Bax are markers of apoptosis, not necrosis.

Validity of the findings

-

Additional comments

- The authors need to highlight the novelty of the findings more clearly.
- The authors should include a graphical representation of the normalized expression data in the main manuscript.

·

Basic reporting

I suggest a revision of the English language to make the reading and comprehension of the article more fluent. The authors used repeted linkers (At present), and the manuscript contains some spelling mistakes.

There is missing data on References: 8, 20, and 30.

In reference 20, it seems that there are two citations instead of one, (it should be clarified).

Experimental design

The study is well laid out and appears to be an alternative for sentinel lymph node mapping.

The authors compared this procedure with other ones already established in clinical practice. In my opinion, it has some benefits but also some limitations being the absortion time one of them. Taking 12 hours for staining significantly the lymph node, it would be interesting if increasing the dose or concentration of CNPs, the time, by contrast, should decrease.

Validity of the findings

The results derived from this research reveal the feasibility and reproducibility of the assay.

The figures and tables provided show information that clarifies or complements what is described in the text.

The conclusion is a summary of the study results.

Additional comments

Regarding the euthanasia used for the animals, I should recommend another validated method for this species, such as intravenous anesthetic overdose (like pentobarbital), concussion or a blunt blow to the head (only in rabbits weighing less than 5 kg). The 3R principle intends that researchers seek to improve animal welfare by using a painless but effective procedure.

In my opinion, another group with different concentration could be interesting to be done, in order to compare how the concentration affects on the absortion time. It would be interesting to know if higher concentration of the CNPs, the time for a significant staining could be shorter.

---

## Round 0.2 · accepted · Accept

The authors have addressed all of the reviewers' comments and the manuscript is now ready for publication.

Reviewer 1 ·

Basic reporting

-

Experimental design

-

Validity of the findings

-